# Utilization of complementary and alternative medicine (CAM) by women with breast cancer or gynecological cancer

**Anja Stöcker, Anja Mehnert-Theuerkauf, Andreas Hinz, Jochen Ernst** [ID]*

Department of Medical Psychology and Medical Sociology, University Medical Center Leipzig, Leipzig, Germany

* jochen.ernst@medizin.uni-leipzig.de

**Data Availability Statement:** All data files are available from the figshare.com database (DOI: 10.6084/m9.figshare.22651402).

## Abstract

### Background

Complementary and Alternative Medicine (CAM) has become increasingly popular among cancer patients. The prevalence of CAM use differs worldwide and depends on different sociodemographic and medical characteristics. Findings on predictors for CAM use and its benefits for quality of Life (QoL) have been inconclusive in recent studies.

### Objective

The aim of the study was to determine the prevalence and methods of CAM use in breast cancer (BC) and gynecological cancer (GC) patients, to compare CAM users vs. non-CAM users regarding their sociodemographic and medical characteristics, and to investigate the relationship between CAM use and QoL.

### Methods

In a sub-analysis from a multicenter, cross-sectional study in Germany, we examined 1,214 female cancer patients (897 with BC and 317 with GC). We obtained data from self-reports via standardized questionnaires and measured QoL with the EORTC QLQ-C30.

### Results

In total, 565 of the 1,214 patients (46.5%) used some form of CAM. Its use was higher in BC than in GC patients (48.6% vs. 40.7%). In both cancer groups, the users evaluated CAM as being helpful (BC: 60.8%, GC: 59.6%) The most frequently used CAM methods were vitamins/minerals/micronutrients, homeopathy, and mistletoe therapy. CAM users are younger, more highly educated, and financially better off than non-CAM users. They are also characterized by having been diagnosed with cancer a longer time ago, being at more advanced stages of the disease, and experiencing higher distress levels. CAM use is significantly associated with a lower global health status/ QoL in GC patients. Predictors for CAM use are: high level of vocational education, middle or high monthly income, time span since

**Funding:** This study was funded by a grant from the German Cancer Aid (grant no. 107465) within the psychosocial oncology funding priority program. The funders had no role in study design, data collection and analysis, decision to publish, or preparation of the manuscript

**Competing interests:** The authors have declared that no competing interests exist.

diagnosis of more than 12 months, the status of currently ongoing chemotherapy or hormone therapy treatment, and distress.

## Conclusion

Our data support the findings of previous studies and highlight the need to develop a consistent CAM definition with respect to comparability and evaluation of CAM services. More longitudinal studies are desirable to establish viable associations between CAM use and relevant outcomes such as QoL or disease management.

## Background

According to the WHO, global cancer incidence is currently 19 million and is expected to increase to an estimated 30 million in 2040 [1, 2]. The rise in new cancer cases is met by improved survival rates, primarily due to optimized treatment options and early disease detection. These improved survival rates are often accompanied by chronicity of the disease and thus a reduced quality of life (QoL) due to psychological impairment, physical dysfunction, chronic pain, sleep disturbances, and fatigue [3].

Complementary and alternative medical care options have gained much attention in recent years and for the majority of cancer patients, they have become an important component of their medical care [4]. This refers to treatment methods that are understood as complementary to scientifically based methods of medicine and are therefore subsumed under the term complementary medicine. The term CAM (complementary and alternative medicine) encompasses a group of different medical and health-promoting methods, practices and products that are generally not part of conventional medicine [5]. A recent review of 12 studies showed that 25%– 80% of all cancer patients use CAM services as part of their cancer treatment [6].

Utilization depends on sociodemographic factors and medical characteristics. Among breast cancer (BC) patients, relatively high levels of CAM use are found among younger married women with higher levels of education and income, as well as among patients at an advanced disease stage or relapse and those undergoing chemotherapy, radiotherapy, or hormone therapy [7–10]. In another study of 288 BC patients in Korea, it was shown that CAM was more likely to be used by patients who reported a higher symptom burden compared to patients with a lower symptom burden [11].

For gynecological cancer (GC) patients, there are significantly fewer studies, and the results are inconsistent. In a review by Akpunar et al. that included data from 1,978 patients and summarized results obtained in studies from developed countries, there were statistically significant correlations between CAM use and younger age and higher education levels [12]. A study from Turkey with a small sample size of 67 patients [13], as well as a larger study from Thailand [14] with 370 study participants did not find differences in sociodemographic or medical factors. A number of studies [7, 11–17] substantiate the following motives for CAM use in BC and GC patients: an intent to "strengthen the immune system," "reduce cancer therapy-related side effects," "increase physical and mental fitness", and thereby "improve their quality of life", as well as "cure their cancer." Further central motives were: "regaining control over the disease" and the desire to "actively participate in the treatment process" [18].

CAM offerings are broad. According to a classification by the NCCIH (National Center for Complementary and Integrative Health), they include:

1. natural products such as herbs, vitamins, minerals, and supplements,

2. mind-body practices such as yoga, meditation, and acupuncture, and

3. complementary approaches such as Traditional Chinese Medicine (TCM), Ayurveda, and homeopathy [5].

Worldwide, the most popular CAMs among cancer patients in general are natural products of various plant and processed types, as well as vitamins and minerals [6]. The products used depend strongly on the region. In Arab countries, for example, the preferred CAMs tend to be religious in nature, such as prayers and recitation of the Quran [15, 19]. In European countries, vitamins and minerals are most common in BC patients, as well as relaxation methods, homeopathy, and mistletoe therapy [7, 9]. In GC patients, herbal medicine, vitamins and minerals are favored [12]. Several CAMs are often used together or successively [8, 10, 13, 16]. With regard to satisfaction with the CAM offers, a high mean satisfaction (60%) was shown in BC and GC patients [8, 20].

## Association between CAM use and QoL in BC and GC patients

The recording of QoL scores to determine symptom burden is of central importance, especially in cancer patients. Most studies investigating the relationship between CAM use/nonuse and QoL in BC and GC patients show no significant difference in global QoL in both BC [10, 16, 18] and GC patients [14, 21]. However, in many of these studies, significant relationships were found in the QoL scores for various individual items and subscales [14, 18]. For example, a review by Smits et al. (2015) showed a significant improvement in fatigue, as well as sleep quality and physical functioning in the intervention group (CAM use) compared to the control group in GC patients [21]. Few studies to date have shown improved global QoL with CAM use, such as Albabtain et al. observed in BC patients in Saudi Arabia [15]. A review by Cramer et al. (2017) found that yoga improved health-related QoL and reduced fatigue and sleep disturbances among BC patients, though the effects were only short-term [22].

In summary, the studies point to a high prevalence of use of and satisfaction with CAM among BC and GC patients. The interpretation and evaluation of the previous study results is difficult, mainly due to different definitions of CAM methods. Also, the often small sample sizes allow only very limited comparability and generalization of the results [6].

## Aims of this study

The present study investigates the use of CAM services in 1,214 female patients with BC or GC with the following research questions:

1. What is the proportion of CAM users in the two groups (BC and GC), and as how helpful do they rate their CAM use?

2. Which types of CAM services are predominantly used by BC and GC patients?

3. Which socio-demographic and medical variables characterize CAM users and non-users?

4. What is the relationship between CAM use and QoL in BC and GC patients?

## Material and methods

### Participants and procedures

In this sub-sample analysis, we use data from a large, multicenter, epidemiological, cross-sectional study, the methods of which have been described in more detail elsewhere [23, 24]. Patients were recruited while receiving treatment from oncological inpatient clinics at acute care hospitals, specialized outpatient cancer care facilities, and cancer rehabilitation centers across five study centers in Germany (Freiburg, Hamburg, Heidelberg, Leipzig, and Würzburg). Inclusion criteria consisted of being a cancer patient, between the ages of 18–75, with a malignant tumor. Patients across all tumor entities and disease stages were included. All participants provided written informed consent before taking part in the study. Exclusion criteria were: age younger than 18 or older than 75 years, severe cognitive or physical impairment, language barriers. The study complied with the Declaration of Helsinki and was approved by the research ethic committees of the local medical association in each study center.

Of 5,889 eligible patients, 4,020 (68%) agreed to participate in the study (Fig 1). The primary reasons given for non-participation were lack of interest (55%) and symptom burden (33%). In this analysis, we considered only female patients suffering from BC (897 patients) or GC (317 patients, in particular: 79 cervical, 70 endometrial, 131 ovary and fallopian tube, 37 vulva and vagina).

### Measures

Sociodemographic data were collected using a standardized questionnaire. Disease-related characteristics were obtained from medical charts.

**Assessment of CAM.** Participants were classified as CAM users if they had used at least one type of CAM. Use of CAM was captured by the question "Did you use CAM with regard to your cancer (yes/no), and if so, was it helpful (not at all; a little; some; quite a bit; very much)?" CAM users were then asked which types they had used (anthroposophical medicine, Bach-flowers therapy, enzymes, homeopathy, kinesiology, mistletoe therapy, neural therapy, phytotherapy, traditional Chinese medicine/acupuncture, vitamins/minerals/micronutrients, and others).

**Quality of Life.** Patients' QoL was assessed using the cancer-specific EORTC QLQ-C30 (European Organization for Research and Treatment of Cancer quality of life questionnaire Core 30) [25]. This instrument is widely used and has good reliability and validity. It contains 30 items encompassing five functional scales, three symptom scales, a global health status / QoL scale, and six single items. All of the scores and single items range from 0 to 100. A high score for the global health status represents a high QoL, but a high score for a symptom scale or item indicates more problems and higher levels of symptoms. For our analysis, we focused on the three symptom scales (*fatigue*, *nausea/vomiting*, and *pain*), four symptom items (*insomnia*, *appetite loss*, *constipation*, and *diarrhea*), and the global health status / QoL.

**Distress thermometer.** With the Distress Thermometer (DT), we assessed the global level of distress experienced in the past week on an 11-point visual analogue scale ranging from 0 ("no distress") to 10 ("extreme distress"). This instrument is validated for use in the German language with the recommended cut-off $\geq 5$ to detect cancer-related distress [26].

### Statistical analysis

Means and standard deviations (SDs) were used to describe the distribution of quantitative data. Absolute and relative frequencies were used to present qualitative data. We analyzed associations between categorical variables using chi$^2$-test/ ANOVA and independent *t*-test,

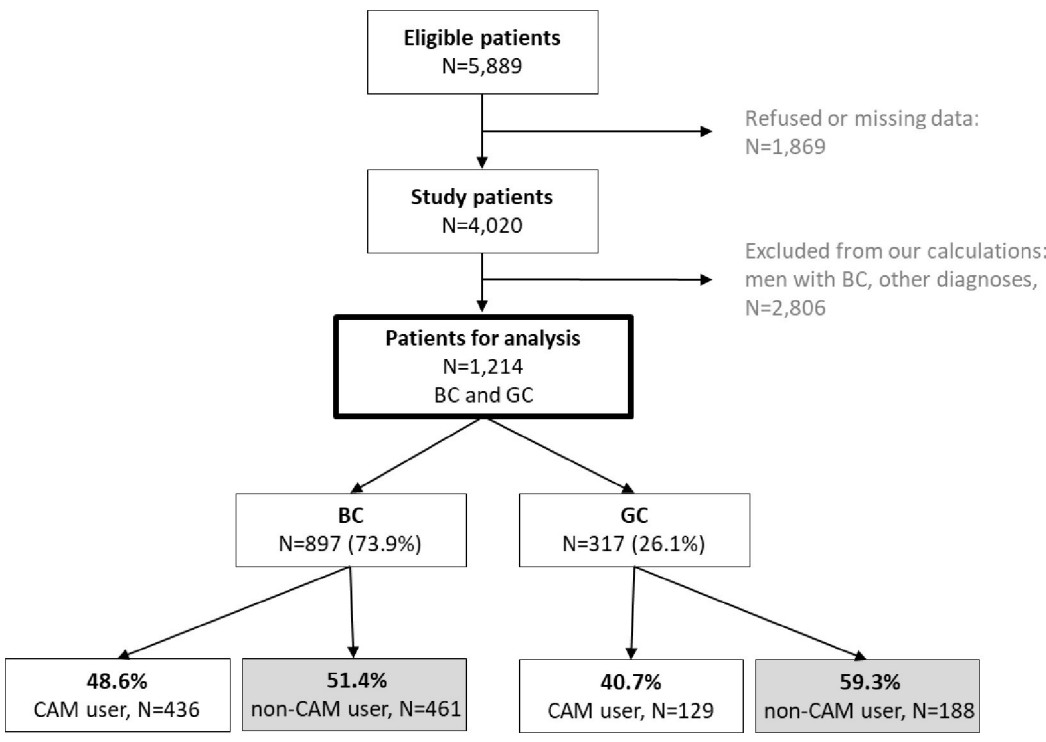

**Fig 1. Flowchart of the sample.**

and for continuous variables we used the Pearson correlation coefficient and Fisher's exact test. To identify predictors for CAM use, binary logistic regression analyzes were performed.

Those sociodemographic and medical factors that showed a significant difference in the bivariate analysis were included in the model. To classify the differences in terms of effect sizes, we used the recommendations by Cohen [27] (d ≥ 0.2 small effect, d ≥ 0.5 medium effect, r ≥ 0.8 large effect). A two-sided $p < 0.05$ was considered statistically significant for all analysis. Statistical analyses were conducted with IBM® SPSS Statistics 26, and Microsoft Excel 2010.

## Results

### Sample characteristics

Fig 1 shows the study enrollment. A subsample of 1,214 female patients were selected, of whom 897 suffered from BC and 317 from GC. Sociodemographic and medical characteristics of BC and GC patients are presented in Table 1, column A. In summary, there were no differences regarding age, vocational education, and monthly income between the cancer groups. By contrast, significant differences were noticed in relationship status (BC 78.3% vs GC 72.3%, p<0.05), occupational situation (BC 49.6% employed, GC 40.8% employed, p<0.003), time since diagnosis (BC 18.2 month vs. GC 11.2 month, p = 0.001), disease stage, past/current treatment, setting, and the distress thermometer (Table 1, column A: sample characteristics).

**Table 1.** Sample characteristics (A) and comparison CAM user vs. non-CAM user (B).

| | (A) Sample characteristics | | | (B) Comparison CAM user vs. non-CAM user | | | |
|---|---|---|---|---|---|---|---|
| | BC N = 897 (73.9%) | GC N = 317 (26.1%) | p-Value [a] | CAM user N = 565 (46.5%) | non-CAM user N = 649 (53.5%) | p-value [a] | Binary logistic regression [b] (Odds Ratio [OR] for CAM use) |
| **Sociodemographic characteristics** | | | | | | | OR and CI |
| **Age** (y) (Mean ± SD) | 55.3±10.9 | 55.5±12.2 | 0.817 | | | | |
| < **50** | 270 (30.8%) | 92 (29.6%) | 0.720 | 197 (35.2%) | 165 (26.3%) | 0.001** | Reference |
| ≥ **50** | 606 (69.2%) | 219 (70.4%) | | 363 (64.8%) | 462 (73.7%) | | 0.77 (0.56–1.05) |
| **Living with a partner** | | | < 0.05* | | | 0.887 | |
| Yes | 641 (78.3%) | 201 (72.3%) | | 415 (76.6%) | 427 (76.9%) | | |
| No | 178 (21.7%) | 77 (27.7%) | | 127 (23.4%) | 128 (23.1%) | | |
| **Vocational education** | | | 0.400 | | | 0.001** | |
| Apprenticeship/vocational school | 505 (59.6%) | 206 (68.7%) | | 327 (58.7%) | 384 (65.0%) | | Reference |
| University of applied sciences/ University | 225 (26.5%) | 58 (19.3%) | | 163 (29.3%) | 120 (20.3%) | | **1.57 (1.09–2.28)** |
| Others | 53 (6.3%) | 17 (5.7%) | | 37 (6.6%) | 33 (5.6%) | | 1.27 (0.63–2.53) |
| No vocational education | 65 (7.7%) | 19 (6.3%) | | 30 (5.4%) | 54 (9.1%) | | 0.84 (0.44–1.59) |
| **Monthly income (€)** | | | 0.102 | | | 0.003** | |
| <1000 | 304 (44.6%) | 109 (48.0%) | | 182 (40.1%) | 231 (50.8%) | | Reference |
| 1000-<2000 | 245 (35.9%) | 88 (38.8%) | | 177 (39.0%) | 156 (34.3%) | | **1.41 (1.01–1.96)** |
| ≥ 2000 | 133 (19.5%) | 30 (13.2%) | | 95 (20.9%) | 68 (14.9%) | | 1.52 (0.96–2.40) |
| **Occupational situation** | | | 0.003** | | | 0.260 | |
| Employed | 420 (49.6%) | 119 (40.8%) | | 284 (51.4%) | 255 (43.5%) | | |
| Unemployed | 142 (16.8%) | 43 (14.7%) | | 85 (15.4%) | 100 (17.1%) | | |
| Retired | 284 (33.6%) | 130 (44.5%) | | 183 (33.2%) | 231 (39.4%) | | |
| **Medical characteristics** | | | | | | | |
| **Time since diagnosis in months,** Mean±SD | 18.2±32.8 | 11.2±24.1 | 0.001** | 21.7±36.9 | 11.4±23.0 | < 0.001*** | |
| ≤ 12 months | 556 (70.2%) | 217 (78.1%) | 0.013* | 327 (63.1%) | 446 (80.8%) | <0.001*** | Reference |
| > 12 months | 236 (29.8%) | 61 (21.9%) | | 191 (36.9%) | 106 (19.2%) | | **2.01 (1.42–2.85)** |
| **Disease stage (UICC)** | | | < 0.001*** | | | 0.035* | |
| 0-I | 284 (31.9%) | 66 (20.8%) | | 145 (25.8%) | 205 (31.8%) | | Reference |
| II | 268 (30.1%) | 25 (7.9%) | | 147 (26.1%) | 146 (22.7%) | | 1.29 (0.84–1.99) |
| III | 96 (10.8%) | 34 (10.7%) | | 64 (11.4%) | 66 (10.2%) | | 1.15 (0.68–1.96) |
| IV | 119 (13.4%) | 74 (23.3%) | | 103 (18.3%) | 90 (14.0%) | | 1.31 (0.79–2.17) |
| Unclear | 123 (13.8%) | 118 (37.2%) | | 104 (18.5%) | 137 (21.3%) | | 1.16 (0.73–1.83) |
| **Past/current treatment:** (multiple answers possible) | | | | | | | |
| Surgery | 795 (91.4%) | 270 (86.0%) | 0.024* | 495 (46.5%) | 570 (53.5%) | 0.560 | |
| Radiotherapy | 598 (68.8%) | 108 (34.5%) | < 0.001*** | 347 (49.2%) | 359 (50.8%) | 0.043* | 1.14 (0.830–1.58) |
| Chemotherapy | 540 (61.9%) | 179 (57.4%) | 0.023* | 387 (53.8%) | 332 (46.2%) | < 0.001*** | **1.66 (1.20–2.30)** |
| Hormone therapy | 344 (39.5%) | 1 (0.3%) | <0.001*** | 188 (54.5%) | 157 (45.5%) | 0.001** | 1.20 (0.84–1.71) |
| Not specified | 117 (13%) | 21 (6.6%) | | 65 (11.5%) | 73 (11.2%) | | |
| **Setting** | | | <0.001*** | | | 0.020* | |
| Inpatient | 223 (24.9%) | 137 (43.2%) | | 152 (26.9%) | 208 (32.0%) | | Reference |
| Outpatient | 442 (49.3%) | 102 (32.2%) | | 249 (44.1%) | 295 (45.5%) | | 1.20 (0.82–1.74) |

*(Continued)*

**Table 1.** (Continued)

| | (A) Sample characteristics | | | (B) Comparison CAM user vs. non-CAM user | | | |
|---|---|---|---|---|---|---|---|
| | BC N = 897 (73.9%) | GC N = 317 (26.1%) | p-Value [a] | CAM user N = 565 (46.5%) | non-CAM user N = 649 (53.5%) | p-value [a] | Binary logistic regression [b] (Odds Ratio [OR] for CAM use) |
| Rehabilitation | 232 (25.9%) | 78 (24.6%) | | 164 (29.0%) | 146 (22.5%) | | **1.58 (1.02–2.46)** |
| **Distress Thermometer** | | | 0.001** | | | <0.001*** | |
| Not distressed, if < 5 points | 405 (48.3%) | 109 (36.9%) | | 212 (38.8%) | 302 (51.4%) | | Reference |
| Distressed, if ≥ 5 points | 434 (51.7%) | 186 (63.1%) | | 335 (61.2%) | 285 (48.6%) | | **1.87 (1.38–2.53)** |

[a] ANOVA, Pearson correlation, Fisher's exact test; significance level:

* $p < 0.05$;

** $p < 0.01$;

*** $p < 0.001$

[b] significant values are highlighted in bold

## Frequency of CAM use in BC and GC patients and perceived benefit of CAM use

Fig 1 shows that, overall, 46.5% of the patients used CAM. As Fig 2 shows, 21.1% of BC patients and 20.2% of GC patients used exactly one type of CAM, and about 1 in 10 women used two types (BC 12.6%, GC 9.8%). 60% of respondents in both disease groups rated the methods as quite a bit or very helpful, and a small proportion of 13% rated them as a little or not helpful. A comparison of the groups indicates that BC patients were significantly more likely to use CAM than patients with GC (48.6% vs 40.7%, p = 0.015). The proportion of non-users of both groups together is 53.5%.

## Predominantly used types of CAM services in BC and GC patients

Fig 3 shows the most frequently used types of CAM in both groups: vitamins/minerals/micro-nutrients, homeopathy, and mistletoe therapy. In contrast to that, Bach-flowers therapy, kinesiology, and neural therapy were hardly used. There was a statistically significant difference in the use of vitamins/minerals/micronutrients between BC patients and GC patients (BC 41.2% vs GC 30.0%, p<0.001) and in the use of enzymes (BC 11.0% vs GC 4.0%, p<0.001).

## Sociodemographic and clinical characteristics of CAM users

Table 1, column B, shows the comparison between CAM users and non-CAM users. A significant difference can be seen in the following characteristics: CAM users are younger (53.8 vs 56.7 years, p<0.001), have higher levels of vocational education (p = 0.001), higher monthly incomes (p = 0.003), more advanced disease stages (p = 0.035), and they were more often distressed (61.2% vs. 48.6%, p<0.001). The time since diagnosis in CAM users was 21.7 months and in non-CAM users 11.4 months (p<0.001.). Among the patients receiving chemotherapy, the proportion of CAM users was higher than among patients without chemotherapy (p<0.001).

## Association between CAM use and QoL in BC and GC patients

Table 2 displays the mean value for each symptom scale, single item, and the global health status / QoL of the EORTC QLQ-C30 questionnaire. The proportion of high symptom severity is

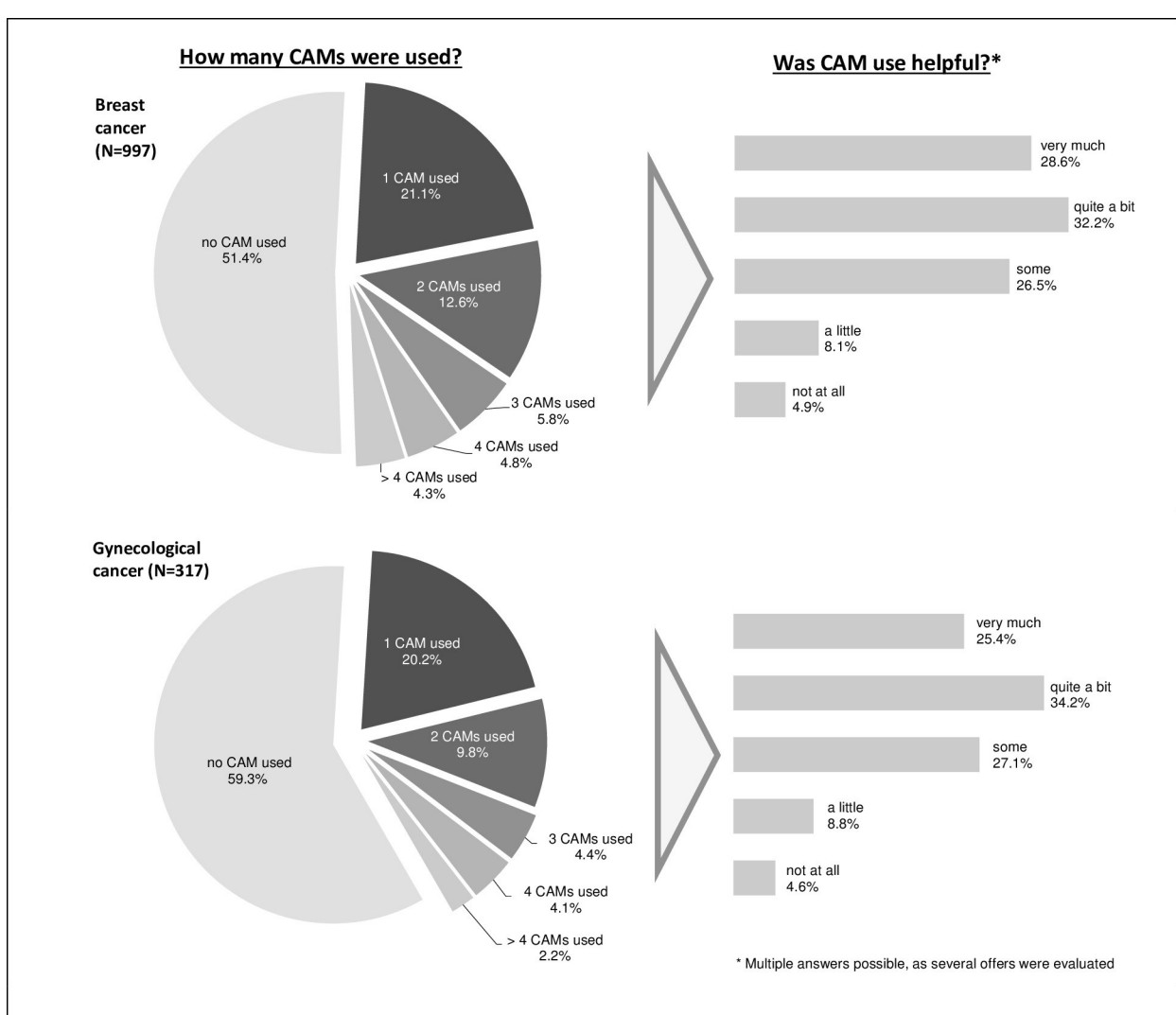

**Fig 2. Prevalence of CAM use in BC and GC patients and its benefit.**

significantly higher among CAM users than among non-CAM users in the dimensions *fatigue* (p = 0.003), *appetite loss* (p = 0.032), and *diarrhea* (p = 0.007) in GC patients, in *pain* in both groups (BC p = 0.018, GC p = 0.032), and *insomnia* in BC patients only (p = 0.007). CAM users have (not significantly) better QoL scores in comparison to non-CAM users for *nausea/vomiting*, *appetite loss* (BC) and *constipation* (GC). The global health status / QoL among GC patients is significantly higher in non-CAM users than in CAM users (54.5 vs 47.9, p = 0.01).

The results of the logistic regression model are shown in Table 1, column B.

The likelihood of using CAM is increased in patients with higher educational status (OR = 1.57), middle income in comparison to low income (OR = 1.41), patients undergoing chemotherapy (OR = 1.66), being treated in a rehabilitation unit (OR = 1.58), and with higher distress levels (OR = 1.87). CAM is most likely to be used if the time since diagnosis is more than 12 months (OR = 2.01).

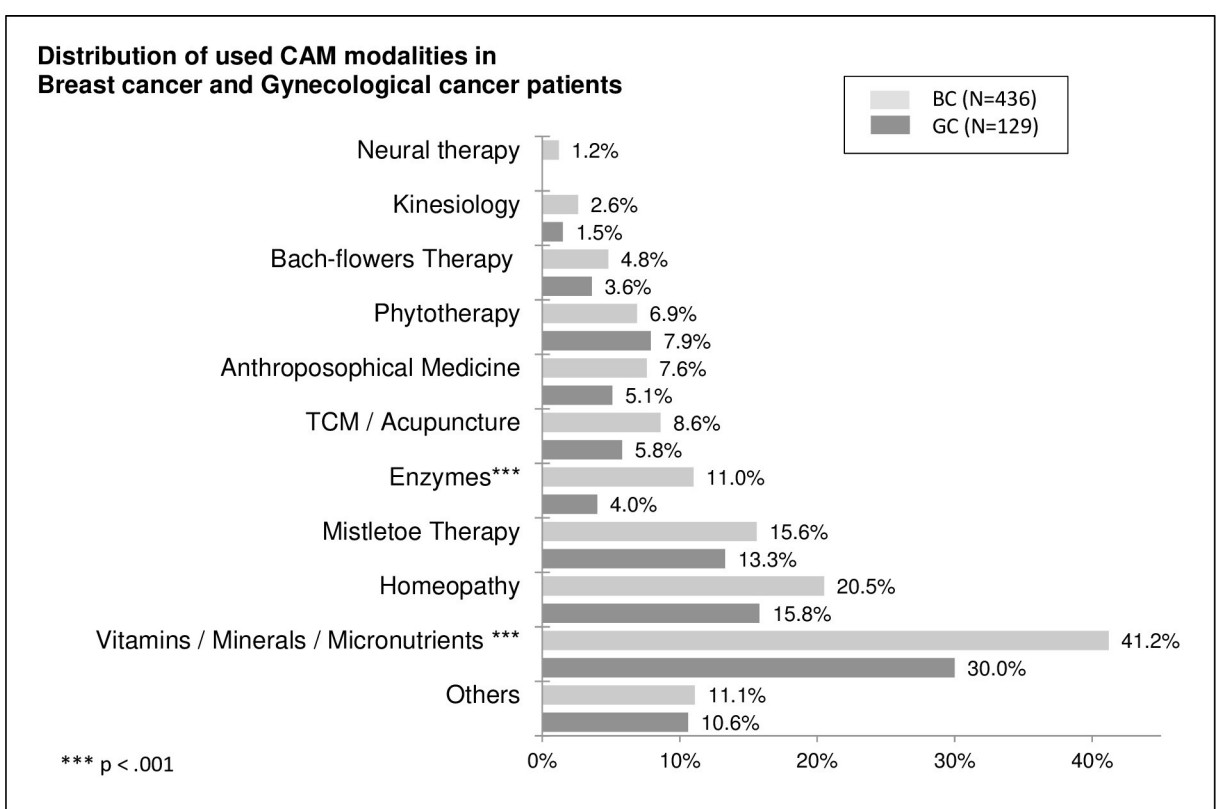

**Fig 3. Distribution of types of CAM among users.**

## Discussion

In this analysis, we surveyed German BC and GC patients concerning their usage of CAM and related associations with QoL.

The prevalence of CAM use in the total study cohort was 46.5%. This is in line with other surveys of BC and GC patients [20, 28, 29]. However, there are several studies with higher prevalence rates, the reasons for which are manifold. In a review report [12] with GC patients from the USA, Europe, Israel, China, and Australia, the prevalence rates varied from 40.3% to 94.7%. According to the authors, the reasons for that broad range are: discrepancies in CAM definitions, diverse CAM modalities, varying research methods (self-report, interviews), as well as geographic and cultural differences in combination with varying attitudes of acceptance of CAM [12].

However, recent studies in particular show an increasing acceptance of CAM methods not only among cancer patients but also among practitioners. CAM is being integrated into conventional therapy, and specialized treatment units are being developed in hospitals in many places, a progression that is leading to increases in the prevalence of CAM use [17, 30, 31]. The difference in prevalence between BC and GC patients in our cohort is statistically significant (BC 48.6% vs GC 40.7%, p<0.015), similar to those observed by Fasching et al. [29] (BC 50.1% vs GC 44.0%, p = 0.058). The authors understood this difference to be due to BC patients belonging to a better organized network and thus having access to more information than GC patients [29]. Paepke et al. also showed a higher CAM use in BC patients (56.7%) compared to GC patients (43.3%), but did not suggest an explanation for why this is the case [32]. They also found that there was a high interest in CAM in the non-user group.

**Table 2.** QoL (EORTC QLQ-C30) scores between CAM users and non-CAM users.

| EORTC | | CAM Users (N = 565) | non-CAM Users (N = 649) | Difference [a] | p [b] | Cohen's \|d\| [c] |
|---|---|---|---|---|---|---|
| | | M ± SD | M ± SD | | | |
| Fatigue | | | | | | |
| | BC | 49.8 ± 29.6 | 46.1 ± 29.6 | 3.7 | 0.067 | 0.125 |
| | GC | 62.2 ± 28.3 | 52.2 ± 28.4 | 10.0 | 0.003** | 0.353 |
| Nausea & Vomiting | | | | | | |
| | BC | 6.9 ± 15.2 | 8.2 ± 18.9 | -1.3 | 0.266 | 0.076 |
| | GC | 17.8 ± 24.9 | 12.8 ± 22.6 | 5.0 | 0.068 | 0.210 |
| Pain | | | | | | |
| | BC | 34.2 ± 33.6 | 28.9 ± 31.9 | 5.3 | 0.018* | 0.162 |
| | GC | 47.5 ± 34.0 | 38.9 ± 34.9 | 8.6 | 0.032* | 0.250 |
| Insomnia | | | | | | |
| | BC | 49.8 ± 36.9 | 43.0 ± 37.6 | 6.8 | 0.007** | 0.183 |
| | GC | 51.6 ± 37.8 | 48.4 ± 35.9 | 3.2 | 0.469 | 0.087 |
| Appetite Loss | | | | | | |
| | BC | 14.7 ± 26.5 | 15.9 ± 27.9 | -1.2 | 0.515 | 0.044 |
| | GC | 30.2 ± 35.6 | 21.7 ± 32.6 | 8.5 | 0.032* | 0.249 |
| Constipation | | | | | | |
| | BC | 13.4 ± 27.0 | 10.9 ± 25.1 | 2.5 | 0.163 | 0.096 |
| | GC | 22.1 ± 33.8 | 24.9 ± 37.8 | -2.8 | 0.505 | 0.078 |
| Diarrhea | | | | | | |
| | BC | 11.4 ± 24.2 | 9.4 ± 22.3 | 2.0 | 0.209 | 0.086 |
| | GC | 25.5 ± 36.2 | 15.1 ± 28.9 | 10.4 | 0.007** | 0.318 |
| Global health status / QoL | | | | | | |
| | BC | 59.4 ± 20.9 | 60.9 ± 21.8 | -1.5 | 0.298 | 0.070 |
| | GC | 47.9 ± 22.4 | 54.5 ± 21.4 | -6.6 | 0.010** | 0.301 |

[a] positive score: higher level of CAM user

[b] significance level:

* p < 0.05;

** p < 0.01

[c] Cohen's d: 0.2 small effect, 0.5 medium effect, > 0.8 large effect

In contrast to the well-researched CAM prevalence, there is significantly less evidence on patient satisfaction regarding their CAM use. Our study shows a 60% satisfaction rate with CAM use ("very satisfied" and "satisfied"). These results have also been found at the same level in comparable studies [8, 20].

The most popular CAMs in both groups in our study were (1) vitamins/ minerals/ micronutrients, (2) homeopathy, and (3) mistletoe. These rankings are mostly in line with previous studies [7, 9, 30, 31], though our results reflected lower percentages for the use itself [17]. Neural therapy, kinesiology, and Bach-flowers therapy were hardly used. The CAM types/modalities in the studies from different countries and cultures are as varied as the respective prevalence of their use [6], a fact which limits their comparability.

CAM use is dependent on various sociodemographic and medical characteristics. In our study, the following characteristics were statistically significantly associated with CAM use: younger age (< 50 years), higher vocational education, higher monthly income, longer time since diagnosis, more advanced disease stage, treatment with radiotherapy, chemotherapy and hormone therapy, clinical setting (rehabilitation), and higher levels of distress. These

associations are also reported in a number of other studies [6–10, 33]. In addition, there are studies that have failed to demonstrate any associations between sociodemographic or medical characteristics of CAM users [13, 14, 34]. Reasons for this are: frequently small sample sizes, differences in the availability of CAM, costs in different cultural settings and national health care systems.

Consistent with previous studies, the following predictors of CAM use we found were: higher education [7, 10, 11, 35], middle or high monthly income [10], time since diagnosis > 12 months [7, 8, 11].

However, other medical characteristics such as past/current treatment with chemotherapy, in-patient status in a rehabilitation setting, and higher distress levels are also associated with a significantly higher probability of CAM use in our study. Age also differs significantly between CAM users and non-CAM users in the bivariate analysis (p = 0.001). This statistical effect, however, is no longer detectable in the regression analysis, once it is influenced by the parameters of vocational education and distress level. Numerous other studies have demonstrated age as a predictor of CAM [7, 18, 28, 35]. Furthermore, in contrast to our results, other studies have also shown employment status [8] and stage of the disease [10, 18] to be predictors of CAM use. Often, however, no predictors of CAM use could be found [31, 36]. The CAM users in our study showed significantly lower QoL scores in many areas, i.e. a higher symptom burden compared to the non-users. In particular, these include *pain* in BC and GC patients, *insomnia* in BC patients, and *fatigue*, *appetite loss*, and *diarrhea* in GC patients. From a clinical point of view, this could mean that the motive for CAM use is often hope for symptom relief and to have an active way of coping with the disease. This is also supported by the results of Hwang et al., who showed more frequent CAM use in BC patients with greater symptom burden [11]. The global health status / QoL of GC patients was significantly higher in non-CAM users (54.5 vs 47.9, p = 0.01) in our study.

Due to the cross-sectional study design, no causal relationships can be derived here. Ben-Arye et al. examined CAM use at two different time points (baseline and a 6 to 12-week follow-up) and showed that most outcomes improved in the CAM intervention group [37]. Klafke et al. [38] were able to demonstrate improved QoL scores in *fatigue*, *emotional functioning*, and *dyspnea* among BC and GC CAM users at the end of chemotherapy compared to the start of chemotherapy. For BC patients in our study, no difference in global QoL was found between CAM users and non-CAM users, which is consistent with the results of most previous studies [10, 14, 16, 18, 21].

What could these findings mean for clinical practice?

Patients with breast and gynecological cancers have a broadly common clinical context with regard to diagnostics, therapy and after-care and are mostly treated by the same specialist. Therefore, both cancer types are discussed within the same framework.

The focus on a large patient group within our study demonstrates to which extent CAM methods are used and which differences can be found between the two types of cancer patients. For doctors as well as for clinical staff, it is very important to know which preferences patients have regarding their aftercare within the framework of oncological therapy. It helps all physicians to define and recommend the right additional medical treatment for each patient individually, which can be easily integrated into clinical care plans.

Although the number of studies on CAM use has multiplied over the last 10 years, evidence for the effects of CAM is still very sparse. Above all, longitudinal studies are necessary in order to evaluate causalities and also to derive better dose-response relationships. Adverse effects as well QoL scores respectively should be investigated regularly before, during and after the cancer therapy, preferably by using standardized questionnaires such as the EORTC QLQ-C30 or the QLQ-BR23. This approach would enable the evaluation of and recommendations for the

use of CAM methods at a very individualized level. In particular, for the most popular CAM group of vitamins/minerals/micronutrients there are no dose specifications and the application should ideally be monitored by taking blood samples. Our and many other studies have shown that older patients in particular are less likely to use CAM [9, 12], yet high BMI and concomitant medication are also associated with a lower interest in integrative medicine. [28]. It represents a vicious circle as old age and high BMI increase the risk of chronic diseases and cancer itself. Furthermore, there a no effective screening strategies for gynecological malignancies such as Ovarian or endometrial cancer as shown in a recent study by Giannini et al [39]. CAM is to be considered here in terms of tertiary prevention in order to achieve the best possible QoL. The population will continue to age in the future, so that an increasing number of older people with comorbidities will be diagnosed with cancer compared to today. CAM use should also enable a strategy to improve the oncological outcome comparable to the preoperative evaluation of the frailty of gynecological cancer patients proposed by D'Oria et al [40].

## Limitations and strengths of the study

There were some limitations of our study. First, it was a cross-sectional design and, therefore, no conclusion regarding causality with CAM use can be drawn. Second, we did not ask about participants' motivation for CAM use and non-CAM use, and we did not ask about any previous patterns of CAM use prior to the respondent's cancer diagnosis.

The strengths of this study are its large sample size and the multicenter study design.

By including patients in different treatment settings (acute care clinic, outpatient departments, rehabilitation centers), it is possible to examine CAM use throughout the treatment course. Because the methodology is identical, is it possible to directly compare both cancer types.

## Conclusion

Though this is not the first study to examine patients with gynecological malignancies and their use of CAM (prevalences, modalities, characteristics, predictors, and associations with QoL), it does underline previous study results, especially in Germany. Creating uniform definitions for CAM and CAM modalities with regard to the comparability of study results is of enormous importance for further research in this area. Likewise, larger longitudinal studies should be conducted in the future to capture causality between CAM use and QoL.

### Summary

About half of our study cohort uses CAM, BC more than GC patients, and users are predominantly satisfied with this use. There are sociodemographic and medical predictors of CAM use such as younger age, high educational status, no acute care (or in rehabilitation), and higher distress. In GC patients, CAM use is more often associated with worse QoL.

### Acknowledgments

We thank all healthcare teams who were involved in data collection across all of the local study centers.

### Author Contributions

**Conceptualization:** Anja Mehnert-Theuerkauf, Jochen Ernst.

**Data curation:** Anja Stöcker, Jochen Ernst.

**Formal analysis:** Anja Stöcker, Andreas Hinz, Jochen Ernst.

**Funding acquisition:** Anja Mehnert-Theuerkauf.

**Investigation:** Anja Mehnert-Theuerkauf.

**Methodology:** Andreas Hinz.

**Writing – original draft:** Anja Stöcker, Anja Mehnert-Theuerkauf, Andreas Hinz, Jochen Ernst.

**Writing – review & editing:** Anja Stöcker, Anja Mehnert-Theuerkauf, Andreas Hinz, Jochen Ernst.

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
