## [Decision Letter · Decision Letter 0]

21 Feb 2023

PONE-D-23-00256Utilization of Complementary and Alternative Medicine (CAM) by Women with Breast Cancer or Gynecological CancerPLOS ONE

Dear Dr. Ernst,

Thank you for submitting your manuscript to PLOS ONE. After careful consideration, we feel that it has merit but does not fully meet PLOS ONE’s publication criteria as it currently stands. Therefore, we invite you to submit a revised version of the manuscript that addresses the points raised during the review process.

We look forward to receiving your revised manuscript.

Kind regards,

Antonio Simone Laganà, M.D., Ph.D.

Academic Editor

PLOS ONE

Journal Requirements:

"This study was funded by a grant from the German Cancer Aid (grant no. 107465) within the psychosocial oncology funding priority program."

Additional Editor Comments:

The topic of the manuscript is interesting. Nevertheless, the reviewers raised several concerns: considering this point, I invite authors to perform the required major revisions.

Reviewers' comments:

Reviewer's Responses to Questions

**Comments to the Author**

1. Is the manuscript technically sound, and do the data support the conclusions?

Reviewer #1: Yes

Reviewer #2: Yes

Reviewer #3: Partly

Reviewer #4: Yes

Reviewer #5: Partly

2. Has the statistical analysis been performed appropriately and rigorously? 

Reviewer #1: Yes

Reviewer #2: Yes

Reviewer #3: N/A

Reviewer #4: Yes

Reviewer #5: I Don't Know

3. Have the authors made all data underlying the findings in their manuscript fully available?

Reviewer #1: Yes

Reviewer #2: Yes

Reviewer #3: No

Reviewer #4: Yes

Reviewer #5: Yes

4. Is the manuscript presented in an intelligible fashion and written in standard English?

Reviewer #1: No

Reviewer #2: Yes

Reviewer #3: No

Reviewer #4: Yes

Reviewer #5: Yes

5. Review Comments to the Author

Reviewer #1: I read with great interest the Manuscript titled "Utilization of Complementary and Alternative Medicine (CAM) by Women with Breast Cancer or Gynecological Cancer " which falls within the aim of the Journal.

In my honest opinion, the topic is interesting enough to attract the readers’ attention.

Nevertheless, authors should clarify some point and improve the discussion citing relevant and novel key articles about the topic.

- The whole text should be corrected by a native English speaker in order to make the work clearer and more readable.

-The introduction should be extended and completed. I find interesting a reference to:

D’Oria O., Golia D’Augè T., Baiocco E., Vincenzoni C., Mancini A, Bruno V., …A. Giannini The role of preoperative frailty assessment in patients affected by gynecological cancer: a narrative review Ital J Gynaecol Obstet. 2022 June p.p. 76-83 doi: 10.36129/jog.2022.34.

- Inclusion/exclusion criteria should be better clarified by extending their description.

- Discussions can be expanded and improved by citing relevant articles (I suggest authors to read and insert in references the following article PMID: 36141217).

Considered all these points, I think it could be of interest for the readers and, in my opinion, it deserves the priority to be published after minor revisions.

Reviewer #2: The manuscript is perfectly fine, you may proceed. The manuscript needs no further correction on any ground, as per my opinion, in order to be published in the journal, unless as mentioned by other reviewers.

Reviewer #3: The aim of this study is to investigate the use of CAM services in female patients with BC or GC. It is interesting to discuss the two types of cancer. However, the author did not tell the readers how the two are related. Why do they need to be discussed and compared together? What is the difference in socioeconomic status or background between the two? Even if it is understood that the quality of life of the two is related to the use of alternative medical care, does the author have any suggestions for medical providers to refer to? Overall, the structure of the entire article is loose, and the focus of discussion cannot be found.

Reviewer #4: I read with great interest the Manuscript titled “Utilization of Complementary and Alternative Medicine (CAM) by Women with Breast Cancer or Gynecological CancerUtilization of Complementary and Alternative Medicine (CAM) by Women with Breast Cancer or Gynecological Cancer", which falls within the aim of this Journal.

In my honest opinion, the topic is interesting enough to attract the readers’ attention. Methodology is accurate and conclusions are supported by the data analysis. Nevertheless, authors should clarify some point and improve the discussion citing relevant and novel key articles about the topic.

Authors should consider the following recommendations:

- Manuscript should be further revised by a native English speaker

- Inclusion criteria should be better clarified

- What are the actual clinical implications of this study? it is important to report the results obtained by the authors in the context of clinical practice and to adequately highlight what contribution this study adds to the literature already existing on the topic and to future study perspectives

- Does this manuscript conform the Enhancing the QUAlity and Transparency Of health Research (EQUATOR) network guidelines? It would be mandatory to declare about this element

- Was this study registered? I could not find any information about this point.

In light of the advanced techniques to detect early-stage disease, to date it is mandatory to consider even the possibility of fertility-sparing approaches in order to preserve reproductive potential of women affected by gynecological cancers. I invite authors to discuss this point, referring to: PMID: 22398708; PMID: 34769256.

- Accumulating evidence suggests that obesity and metabolic diseases may play a key role in increasing the risk of cancers, modulating pivotal cross-talk pathways for cell proliferation and differentiation. I recommend to stress these important points about prevention and screening strategies in gynecological oncology, referring to: PMID: 35314087; PMID: 36141217.

Reviewer #5: Dear Authors,

This study is very interesting albeit limited, as it is a cross-sectional design and therefore, no conclusion can be drawn regarding causality with the use of CAM. including patients in different therapeutic contexts (intensive care clinic, outpatient clinics, rehabilitation centers), it is not possible to examine the use of CAM in a specific way, but the result has some bias due to the variables. While it must be recognized that since the methodology is identical, both types of cancer can be directly compared. I would have appreciated evaluations of predictive outcomes for the use of CAM, and its benefits for QoL.

Finally, for a multidisciplinary approach we propose to mention:

DOI:10.3390/cancers14143457

DOI:10.3390/cancers14143457

6. PLOS authors have the option to publish the peer review history of their article (what does this mean?). If published, this will include your full peer review and any attached files.

Reviewer #1: **Yes: **Tullio Golia D'Augè

Reviewer #2: **Yes: **Chaitali Nath

Reviewer #3: No

Reviewer #4: No

Reviewer #5: No

---

## [Author Response · Author response to Decision Letter 0]

20 Apr 2023

Dear Reviewer,

Thank you very much for your helpful comments and advice.

The responses to the reviewers are uploaded in an extra file.

---

## [Decision Letter · Decision Letter 1]

2 May 2023

Utilization of Complementary and Alternative Medicine (CAM) by Women with Breast Cancer or Gynecological Cancer

PONE-D-23-00256R1

Dear Dr. Ernst,

We’re pleased to inform you that your manuscript has been judged scientifically suitable for publication and will be formally accepted for publication once it meets all outstanding technical requirements.

Kind regards,

Antonio Simone Laganà, M.D., Ph.D.

Academic Editor

PLOS ONE

Additional Editor Comments (optional):

The authors performed the required corrections, which were positively evaluated by the reviewers. I am pleased to accept this paper for publication.

Reviewers' comments:

Reviewer's Responses to Questions

**Comments to the Author**

1. If the authors have adequately addressed your comments raised in a previous round of review and you feel that this manuscript is now acceptable for publication, you may indicate that here to bypass the “Comments to the Author” section, enter your conflict of interest statement in the “Confidential to Editor” section, and submit your "Accept" recommendation.

Reviewer #1: All comments have been addressed

Reviewer #2: All comments have been addressed

2. Is the manuscript technically sound, and do the data support the conclusions?

Reviewer #1: Yes

Reviewer #2: Yes

3. Has the statistical analysis been performed appropriately and rigorously? 

Reviewer #1: Yes

Reviewer #2: N/A

4. Have the authors made all data underlying the findings in their manuscript fully available?

Reviewer #1: Yes

Reviewer #2: Yes

5. Is the manuscript presented in an intelligible fashion and written in standard English?

Reviewer #1: Yes

Reviewer #2: Yes

6. Review Comments to the Author

Reviewer #1: I read your work with great interest and pleasure. The work with the changes made after my advices and those of the other reviewers is complete and, in my opinion, ready for publication.

Reviewer #2: The manuscript is well written. There are no major revisions required for the manuscript to be published.

7. PLOS authors have the option to publish the peer review history of their article (what does this mean?). If published, this will include your full peer review and any attached files.

Reviewer #1: **Yes: **Tullio Golia D'Augè

Reviewer #2: **Yes: **Chaitali Nath

---

## [Editor Report · Acceptance letter]

5 May 2023

PONE-D-23-00256R1 

Utilization of Complementary and Alternative Medicine (CAM) by Women with Breast Cancer or Gynecological Cancer 

Dear Dr. Ernst:

I'm pleased to inform you that your manuscript has been deemed suitable for publication in PLOS ONE. Congratulations! Your manuscript is now with our production department. 

Kind regards, 

on behalf of

Dr. Antonio Simone Laganà 

Academic Editor

PLOS ONE